

# GeneCompete: an integrative tool of a novel union algorithm with various ranking techniques for multiple gene expression data

Panisa Janyasupab[1], Apichat Suratanee[2,3] and Kitiporn Plaimas[1,4]

[1] Department of Mathematics and Computer Science/Faculty of Science, Chulalongkorn University, Bangkok, Thailand
[2] Department of Mathematics/Faculty of Applied Science, King Mongkut's University of Technology North Bangkok, Bangkok, Thailand
[3] Intelligent and Nonlinear Dynamics Innovations Research Center, Science and Technology Research Institute, King Mongkut's University of Technology North Bangkok, Bangkok, Thailand
[4] Omics Science and Bioinformatics Center/Faculty of Science, Chulalongkorn University, Bangkok, Thailand

Corresponding author
Kitiporn Plaimas,
kitiporn.p@chula.ac.th

## ABSTRACT

**Background:** Identifying the genes responsible for diseases requires precise prioritization of significant genes. Gene expression analysis enables differentiation between gene expressions in disease and normal samples. Increasing the number of high-quality samples enhances the strength of evidence regarding gene involvement in diseases. This process has led to the discovery of disease biomarkers through the collection of diverse gene expression data.

**Methods:** This study presents GeneCompete, a web-based tool that integrates gene expression data from multiple platforms and experiments to identify the most promising biomarkers. GeneCompete incorporates a novel union strategy and eight well-established ranking methods, including Win-Loss, Massey, Colley, Keener, Elo, Markov, PageRank, and Bi-directional PageRank algorithms, to prioritize genes across multiple gene expression datasets. Each gene in the competition is assigned a score based on log-fold change values, and significant genes are determined as winners.

**Results:** We tested the tool on the expression datasets of Hypertrophic cardiomyopathy (HCM) and the datasets from Microarray Quality Control (MAQC) project, which include both microarray and RNA-Sequencing techniques. The results demonstrate that all ranking scores have more power to predict new occurrence datasets than the classical method. Moreover, the PageRank method with a union strategy delivers the best performance for both up-regulated and down-regulated genes. Furthermore, the top-ranking genes exhibit a strong association with the disease. For MAQC, the two-sides ranking score shows a high relationship with TaqMan validation set in all log-fold change thresholds.

**Conclusion:** GeneCompete is a powerful web-based tool that revolutionizes the identification of disease-causing genes through the integration of gene expression data from multiple platforms and experiments.

# INTRODUCTION

The identification and examination of differentially expressed genes (DEGs) have become a pivotal foundation for understanding the functioning of genes and their implications in various biological processes and diseases in the dynamic field of genomics and molecular biology. DEGs are genes that exhibit significant changes in activity under various conditions and studying them provides valuable insights into cellular responses, organism development, and the emergence of diseases. Moreover, DEG analysis holds immense potential in shaping the future of biological and medical research. It provides a comprehensive platform for studying and combining diverse datasets, facilitating the unraveling of the complexities of gene activity. DEG research increasingly emphasizes the integration of data from multiple sources. The substantial contributions and promising potential of the data integration in DEG studies are propelling advancements in genomics and molecular biology. This advancement is crucial for advancing our understanding of the intricate molecular networks that govern life processes.

Gene expression analysis allows for a direct assessment of gene expression levels in disease cells compared to control cells. Many algorithms have been developed to identify DEGs. For instance, a combination of the minimum redundancy maximum relevance (mRMR) and shortest path method was employed to identify pancreatic cancer biomarkers (*Shen, Gui & Ma, 2017*). NETBAGs utilized gene expression profiles and protein-protein interactions with network propagation techniques for cancer subtyping or grouping of genes (*Wu et al., 2015*). Additionally, significant genes have been identified by integrating cancer gene expression profiles with somatic mutations (*Di Nanni et al., 2020*). These approaches showcase the diverse range of algorithms and methodologies employed in the identification of DEGs and biomarkers in different diseases.

The development of technology has led to the increasing availability of gene expression data, and the inclusion of a greater number of datasets further reinforces the significance of genes in relation to diseases. Several studies have focused on integrating multiple gene expression data sources (*Borisov & Buzdin, 2022*), exemplified by the identification of key genes associated with prostate cancer using four microarray datasets (*Khan et al., 2022*). By leveraging the combined information from diverse datasets, these studies aim to enhance our understanding of disease-related genes and uncover valuable insights into the molecular mechanisms underlying specific conditions.

RNA sequencing (RNA-Seq) and microarray are two well-known experimental techniques used for gene expression profiling. Each of these experiments has different advantages and limitations. Microarray is a hybridization technique, whereas RNA-Seq is referred to as a sequencing-based technique. Microarray is cost-effective, which is beneficial when dealing with a large number of samples. However, RNA-Seq is increasingly popular as it offers a higher dynamic range and the ability to discover new genes. The

combination of these two techniques allows for a higher number of samples and experiments, leading to the confirmation of gene importance. Combining RNA-Seq and microarray data in gene expression analysis has its strengths and weaknesses, and the rationale for doing so depends on the specific goals of the analysis. RNA-Seq and microarray technologies capture gene expression data differently. RNA-Seq provides more comprehensive and accurate measurements of gene expression, including quantification of novel transcripts and detection of low-abundance genes. In contrast, microarrays are cost-effective and can provide data for a larger number of samples. Combining both yields a broader gene expression picture and enhances validation. Consistent results between RNA-Seq and microarray data boost confidence, reducing false positives and improving reliability. However, integrating data from different platforms requires careful preprocessing and normalization due to differences in sensitivity and dynamic range. Both technologies have their own sources of technical and biological variability, complicating signal identification. Researchers often choose to combine these data sources when studying a complex biological system or when comprehensive gene expression profiling is essential, combining data sources can provide a more complete picture. Combining data from multiple platforms can help validate findings, improving the reliability and robustness of the analysis.

The combining approach has been employed in various research works focusing on different disease, such as pancreatic cancer (*Nisar et al., 2021*), skin cancer (*Gálvez et al., 2019*), and hypertrophic cardiomyopathy (HCM) (*Xu, Liu & Dai, 2021*). The integration of data from multiple sources is crucial for obtaining accurate and reliable biomarkers. Several frameworks have been developed for data integration purposes. Conventional integration techniques mainly involve combining all identified DEGs from different experiments by either taking intersection or union approaches. However, ranking techniques can be a better option for prioritizing genes. RankerGUI applies rank-based statistics to generate ranked profiles and merge them together (*Thind, Tripathi & Guarracino, 2019*). Unlike intersection and union approaches, ranking techniques retain a larger set of important genes. Preprocessing data *via* normalizing expression values of multiple profiles was introduced as a vital tool, namely Rank-In algorithm. This algorithm is referred to as a cross-platform normalization method that minimizes profiling variations (*Tang et al., 2021*). However, while the harmonization algorithm effectively removes batch effects, it can be time-consuming when combining new occurrence datasets. These approaches contribute to the development of robust techniques that enhance the accuracy and effectiveness of biomarker discovery through the integration of data from RNA-Seq and microarray experiments.

Under the same objectives, the integration of several expression datasets would yield a more precise and accurate identification of disease genes. To address the limitation of the existing methods, we propose an integrative web-based tool, namely GeneCompete, which allows all genes from different data sets to compete with each other to be the winner of the diseases (across all experiments). The competition can be formulated based on various ranking methods derived from the results of each experimental dataset, whether from microarray or RNA-seq analyses. While GeneCompete is primarily designed for the

integration of RNA-seq and microarray data, its ranking methods can also be applied to any scenario involving gene ranking. This versatility allows a wider array of datasets and applications to exploit GeneCompete for gene prioritization and ranking, competing against scores derived from various other analyses.

In GeneCompete, several ranking methods have been developed based on the simple winning percentage approach. The rating percentage index (*Pickle & Howard, 1981*) takes into account the winning percentage of opponents. Different approaches are suitable for different applications. For instance, Keener's method (*Keener, 1993*) is designed for ranking football players, while the PageRank technique (*Brin & Page, 1998*) is employed for ranking webpages. In general, ranking methods are used to prioritize a collection of competitors according to their significance level or rating scores. *Langville & Meyer (2012)* compiled a comprehensive array of rating methods. Furthermore, a straightforward forward-looking approach (*Ochieng, London & Krész, 2022*) has been introduced to compare the predictive capabilities of these rating methods. More recently, bi-directional PageRank has improved upon the original PageRank by incorporating additional information about lost games (*Zhou et al., 2022*).

These ranking algorithms have often been applied in sports, and they have the potential to evaluate other domains, such as movies, restaurants, and hotel ratings. Moreover, in biological studies, previous work (*Janyasupab, Suratanee & Plaimas, 2022*) introduced ranking methods for HCM gene expression. Therefore, in this study, our GeneCompete applies these rating techniques to rank genes across various gene expression datasets with a novel concept that considers genes as players or teams in games, and the combination of different datasets is considered as matches in game competitions.

## MATERIALS AND METHODS

This section explains the differential expression analysis, data integration strategies, the web-based platform, ranking methods, and validation techniques, and the gene expression data used in this work.

### Gene expression analysis

Differential expression analysis can be performed in various ways based on the raw gene expression profiling (*Baik, Yoon & Nam, 2020*). In this study, we utilized linear models for microarray and RNA-seq data using the limma package (*Ritchie et al., 2015*). First, we use "GEOquery" package (*Barrett et al., 2012*) to obtain the gene expression profile from Gene Expression Omnibus (GEO) database. Next, we employed the 'lmFit' function to estimate the mean expression levels of disease and normal samples. Following this, the 'contrasts.fit' function was applied to identify the probes that exhibited differential expression between the two types of tissue. Then, we used the empirical Bayes variance moderation method ('eBayes' function) to calculate moderated t-statistics. Lastly, we used the 'toptable' function to extract a table containing the top-ranked probes sorted by $p$-value. It should be noted that probes were converted to gene symbols using 'org.Hs.eg.db' library in R. In cases of duplication, the gene with the lowest $p$-value was chosen. The statistical

information of genes includes log-fold change (*logFC*), *p*-value (*pval.*), and adjusted *p*-value (*adj.pval.*). To differentiate the expression of two groups, *logFC* is defined as

$$logFC = log_2 \left( \frac{x_{disease}}{x_{control}} \right) \qquad (1)$$

where $x_{disease}$ and $x_{control}$ represent the mean gene expression levels in disease and control samples, respectively. A higher *logFC* value indicates higher expression in disease samples compared to normal samples, and conversely for a lower value. The null hypothesis states that there is no significant difference between the averages of the two sample types. Assuming the null hypothesis is true, the *p*-value represents the probability of erroneously rejecting the null hypothesis. Consequently, a *p*-value closer to 0 suggests that the observed difference between the two groups is unlikely to occur due to random chance. To mitigate the risk of false discoveries due to multiple testing, the adjusted *p*-value (*adj.pval*) was computed using the Benjamini-Hochberg correction method. After data collection, the analysis was performed on all datasets. To easily obtain the differential expression table for microarray data, the GEO2R tool is available at https://www.ncbi.nlm.nih.gov/geo/geo2r.

## Data integration and gene expression ranking strategy

As previously mentioned, the method of performing differential expression analysis can vary depending on the suitability of the experimental types. The outcomes of gene expression analysis from various datasets can be likened to 'matches' in a gene competition. Consequently, ranking methods that aim to compute and consolidate scores to determine a competition winner can assist in distinguishing these outcomes. Applying *k* different datasets suggests that we effectively have *k* matches involving all genes in the competition. In our scenario, the log fold change served as a competitive score for each gene. The algorithm of ranking analysis with the conventional data integration method is illustrated in Algorithm 1. This algorithm requires two inputs, *i.e.*, a list of data frames of genes with *logFC* column and row names of gene names, and regulation cases (up-regulation or down-regulation). Common genes are integrated from all datasets and each gene is treated as a player in the ranking model, with the log-fold change used for comparison between two players. A gene with a higher log-fold change is the winner in the up-regulation case while a gene with a lower log-fold change gene is the winner in the down-regulation case. All pairs of genes play an equal number of matches, which is the number of input datasets. The two outputs of the algorithm are the win and loss matrix, which will be further applied in the ranking algorithms. The winning matrix $W = w_{ij}$ represents the total number of matches player *i* wins against player *j*. The losing matrix $L = l_{ij}$ represents the total number of matches player *i* loses against player *j*. However, this intersection process for data integration may eliminate some important genes that are not presented in all datasets.

We further investigated a new union strategy. The sets of genes from all datasets were aggregated together, and the combined genes were separated into two categories: positive and negative *logFC* genes. The process is demonstrated in Fig. 1, and its algorithm is shown in Algorithm 2. This algorithm requires three inputs, with an additional input from the

---

**Algorithm 1  Intersection algorithm**

    **Input:** Table = List of data frames of genes with logFC column and row names of gene names

          Reg = Regulation (Up-regulation or Down-regulation)

    **Output**: W_matrix = A matrix of the winning score of gene i when competing with gene j

          L_matrix = A matrix of the losing score of gene i when competing with gene j

1   T_list ← List of row names of T for all T in Table

2   N_table ← LEN(Table) // Number of input datasets

3   Intersect_set ← T_list[0]

4   **for** k ← 1 **to** N_table-1 **do**

5      Intersect_set ← Intersect_set ∩ T_list[k]

6   **end for**

7   N ← LEN(Intersect_set) // Number of genes in intersection set

8   W_matrix ← $[0]_{N \times N}$

9   L_matrix ← $[0]_{N \times N}$

10  **for** i,j in Intersect_set **do**

11     **for** k ← 1 **to** N_table **do**

12       Dat_fil[k] ← Table[k] with rows of Intersect_set

13      **if** Reg is Up-regulation **then**

14        W[i,j] ← transpose of sign(sign((( Dat_fil[k] ['logFC'])[None,:] - (Dat_fil[k] ['logFC'])[:,None])) + 1)

15      **else if** Reg is Down-regulation **then**

16        W[i,j] ← transpose of sign(sign((( Dat_fil[k] ['logFC'])[:,None ] - ( Dat_fil[k] ['logFC'])[None,:])) + 1)

17      **end if**

18      W[i,i] ← 0

19      L ← |sign(W - 1)|

20      L[i,i] ← 0

21     **end for**

22     W_matrix ← W_matrix + W

23     L_matrix ← L_matrix + L

24  **end for**

25  **return** W_matrix, L_matrix

---

first algorithm being the log-fold change threshold (*thres*). First, the large set of genes is reduced by the condition of *logFC > thres* for up-regulation and *logFC < -thres* for down-regulation. Then, these filtering genes from each dataset are combined and considered as candidates for ranking. The number of games between each pair of genes is determined by the frequency with which the two genes appear together in the same dataset. A gene that exists in a greater number of datasets is likely to participate in a higher number of games. Then, genes similarly compete with *logFC* for each dataset to obtain the win and loss

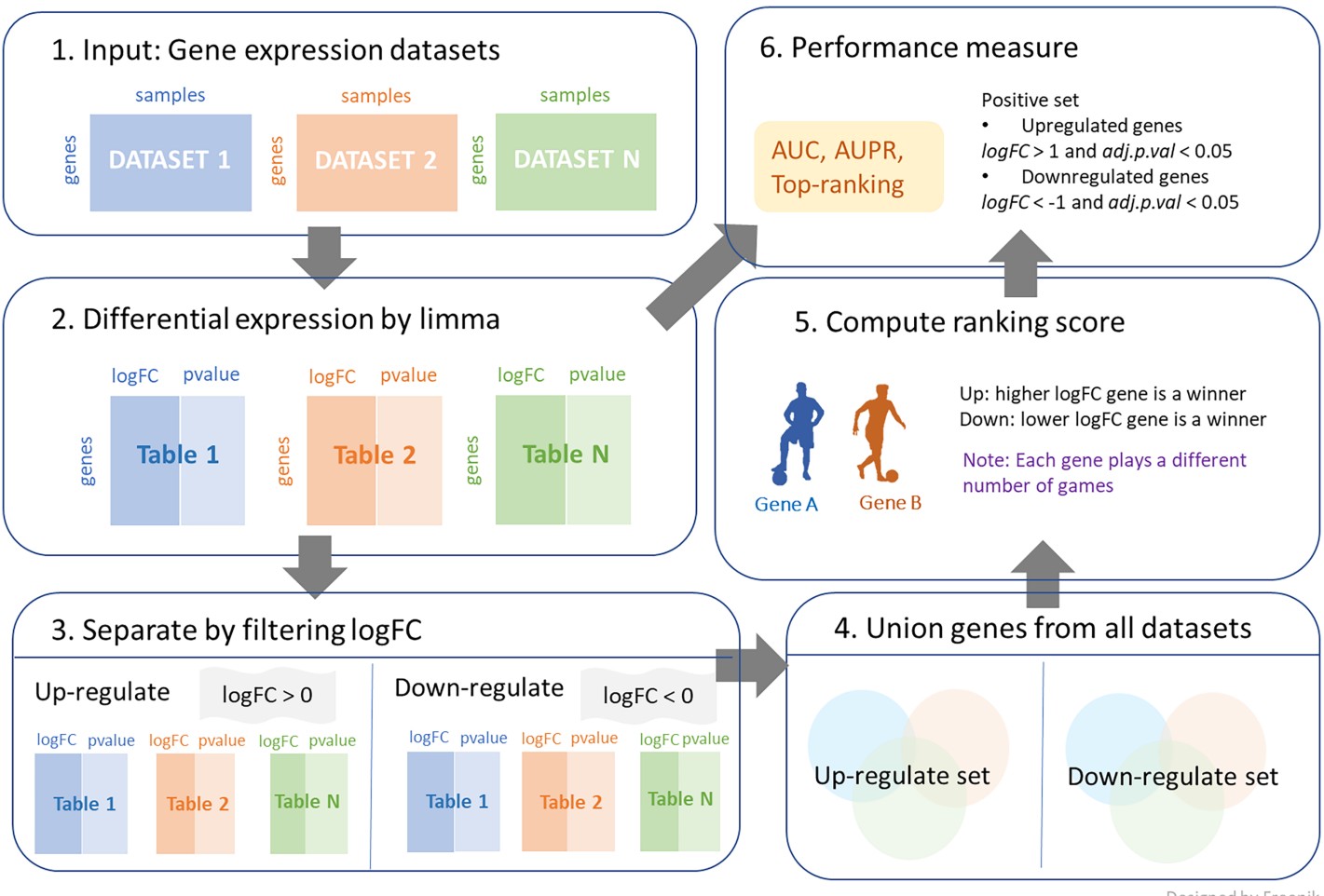

**Figure 1** **The process of integrating multiple gene expression datasets.** First, collecting all gene expression data of interest. Then, calculating *logFC* and *p-value* for differential expressions. Next, filtering *logFC* to identify up-regulated genes and down-regulated genes. After that, performing a union strategy for all datasets of up-regulated set and down-regulated set. Then, computing ranking scores from different ranking techniques. Finally, measuring the performance of the top ranking.

matrix, as output. Note that *thres = 0* is applied in both cases in this work, as shown in Fig. 1.

Both intersection and union integrating processes were applied to multiple gene expression datasets to obtain the ranking scores of genes. The union pipeline may appear similar to the intersection one, but the underlying concepts of the techniques are markedly different. The intersection strategy ranks only genes which are overlapped in all datasets, whereas the union considers all genes as candidates. The main difference between the union and intersection strategies lies in the size of the gene candidate lists they generate. The intersection strategy yields smaller gene candidate lists, potentially missing important candidates that are not present in every dataset. Conversely, the union strategy produces more extensive gene candidate lists, encompassing even rare genes found infrequently in experimental datasets. However, these less common genes may receive lower ranking

**Algorithm 2 Union algorithm**

**Input:** Table = List of data frames of genes with logFC column and row names of gene names

      Reg = Regulation (Up-regulation or Down-regulation)

      thres = Log fold change threshold

**Output:** W_matrix = A matrix of the winning score of gene i when competing with gene j

      L_matrix = A matrix of the losing score of gene i when competing with gene j

1   N_table ← LEN(Table)

2   Data_FC ← [ ]

3   **for** k ← 1 **to** N_table **do**

4     **if** Reg is Up-regulation **then**

5       Data_FC[k] ← List of Table[k] with gene with logFC > thres

6     **else if** Reg is Down-regulation **then**

7       Data_FC[k] ← List of Table[k] with gene with logFC < -thres

8   **end if**

9   **end for**

10   T_list ← List of row names of T for all T in Data_FC

11   Union_set ← { }

12   **for** s in T_list **do**

13     Union_set ← Union_set ∪ s

14   **end for**

15   N ← LEN(Union_set)

16   W_matrix ← $[0]_{N \times N}$

17   L_matrix ← $[0]_{N \times N}$

18   **for** i,j in Union_set **do**

19     **for** k ← 1 **to** N_table **do**

20       Dat_fil[k] ← Table[k] with rows of Union_set ∩ row names of Table[k]

21       Remain[k] ← Union_set - Row names of Dat_fil[k]

22       Matrix_remain[k] ← $[0]_{Remain \times Remain}$

23       **if** Reg is Up-regulation **then**

24         W[i,j] ← transpose of sign(sign((( Dat_fil[k] ['logFC'])[None,:] - ( Dat_fil[k] ['logFC'])[:,None])) + 1)

25       **else if** Reg is Down-regulation **then**

26         W[i,j] ← transpose of sign(sign((( Dat_fil[k] ['logFC'])[:,None] - ( Dat_fil[k] ['logFC'])[None,:])) + 1)

27       **end if**

28       W[i,i] ← 0

29       L ← |sign(W - 1)|

30       L[i,i] ← 0

31       W1 ← merge W and Matrix_remain

32       L1 ← merge L and Matrix_remain

33     **end for**

| Algorithm 2 (continued) | |
|---|---|
| 34 | W_matrix ← W_matrix + W1 |
| 35 | L_matrix ← L_matrix + L1 |
| 36 | **end for** |
| 34 | **return** W_matrix, L_matrix |

scores. To become a top-ranked gene candidate, a gene does not need to participate in every dataset but should perform well in most. The ranking algorithms identify the most potent candidates by evaluating their performance across multiple datasets. The models for competing are presented in the next section, and users can choose the appropriate model for their application.

## Web-based platform for ranking analysis

We have developed an online ranking analysis platform called 'GeneCompete,' which allows users to input a list of gene tables along with their corresponding *logFC* values. The platform generates scores for the genes and ranks them according to the user's selected ranking methods, strategy, and preferred regulation case. Python programming language was used to develop this platform, utilizing a RankIt module to calculate Elo score. If users need to apply the platform for different datasets or diseases, they can access it through https://genecompete.streamlit.app/. For handling large datasets, we also propose the use of a Python function at https://github.com/panisajan/GeneCompete/ (DOI 10.5281/zenodo.8383849).

## Ranking algorithms

This study applies eight ranking algorithms: the win-loss method, Massey's least squares method, Colley's least squares method, Keener's method, Elo's method, Markov method, PageRank method, and Bi-directional PageRank method. These algorithms have traditionally been used to rank sports teams and for various other applications. We apply them to rank genes using multiple gene expression datasets. In this section, we provide the mathematical definitions of the eight ranking methods and clarify the differences between them when using intersection and union strategies. We also explain the formation of ranking competitions and define all the notations used in the models at the beginning of this section.

Assume that there are $k$ gene expression datasets (or $k$ matches in the competition) and $S_t$ is the set of genes in the dataset $t$, where $t = \{1, 2,..., k\}$. Let $S_{int} = S_1 \cap S_2 \cap ... \cap S_k$ be the set of overlapped genes in all datasets. Let $S_t^{up}$ and $S_t^{down}$ be the subset of set $S_t$ with the condition of *logFC > 0* and *logFC < 0* in order. Thus, the sets of union genes after filtering are $S_{up} = \bigcup_{t=1}^{k} S_t^{up}$ and $S_{down} = \bigcup_{t=1}^{k} S_t^{down}$. Consequently, the number of candidate genes is $N = |S_{int}|$ in the intersection pipeline and for union, $N = |S_{up}|$ and $N = |S_{down}|$ in case of up-regulation and down-regulation, respectively.

In the case of $k$ different datasets, we have $k$ rounds of competition with score-based winner selection. For each round, the opponent with a higher *logFC* is the winner and receives a score of 1 in an up-regulation game. In the case of down-regulation, a gene with a lower *logFC* gains a point.

Using the intersection strategies, all overlapped genes play an equal number of games, so the number of matches between gene $i$ and $j$ ($n_{ij}$) is $k$. However, in union strategy, each gene participates in a different number of games, and $n_{ij}$ is based on the number of times gene $i$ and $j$ occur in $S_t^{up}$ and $S_t^{down}$ in case of up-regulation and down-regulation, respectively. Then, the number of games played by gene $i$ can be computed as $N_i = \sum_j n_{ij}$.

The winning matrix $W$ ($w_{ij}$) and losing matrix $L$ ($l_{ij}$) are obtained from Algorithms 1 and 2 for intersection and union strategies, respectively.

### Win-loss method

The win-loss method finds the ratio of the number of wins to the number of matches attended. Let $n_{ij}$ be the number of games played between player $i$ and player $j$, and $w_{ij}$ be the number of times player $i$ wins player $j$. The ranking of player $i$ can be computed as:

$$r_w(i) = \sum_i \frac{w_{ij}}{n_{ij}} \qquad (2)$$

For the intersection strategy, all genes participate in the same of games, with $n_{ij} = k(|S_{int}|-1)$, $\forall i,j$. Then, the ranking can be calculated based on the total number of wins: $r_w(i) = \sum_i w_{ij}$. The maximum value is $k(|S_{int}|-1)$ in the case of winning all players in all matches, and the minimum is 0 when losing every game. In the case of the union strategy, the term $w_{ij}/n_{ij}$ only occurs when player $i$ competes with player $j$. Therefore, the more datasets to which gene $i$ is connected, the more opponents the gene has. Thus, achieving a larger winning percentage with more occurrence in datasets leads to higher gene scores. The advantage of this algorithm lies in its simple concept and low computational time.

### Massey's least squares method

Massey algorithm was originally proposed by *Massey (1997)* to rank football teams. Let $\tau = \sum_i N_i$ and $X$ be the $\tau \times N$ matrix present the outcome of games,

$$X_{ti} = \begin{cases} 1 & \text{if team } i \text{ win } t^{th} \text{ game} \\ -1 & \text{if team } i \text{ lost } t^{th} \text{ game} \\ 0 & \text{otherwise} \end{cases}$$

Each row in $X_{ti}$ contains just two non-zero elements: 1 for the winner and $-1$ for the loser. In the Massey algorithm, explicit opponent indices are not used; instead, it relies on game outcomes (wins and losses) to compute team rankings. This method considers how teams perform against different opponents, ultimately assigning ratings or rankings. While the precise mathematical procedures can differ in various Massey algorithm

implementations, the opponent's identity is typically inferred directly from the game results.

The Massey matrix is defined as $M = X^TX$. It can be expressed in terms of number of games as $M_{ij} = \begin{cases} N_i & if \ i = j \\ -n_{ij} & if \ i \neq j \end{cases}$, where $N_i$ is the number of games played by player $i$. This definition is explained by Massey as $(X^TX)_{ij} = x_i \cdot x_j$, where $x_i = \{1, 2 \ldots, N\}$ is the column vector of $X$. For the diagonal elements, let's consider $x_i \cdot x_i$, in cases where the game is played, $x_i$ can be 1 or −1, resulting in the summation of the number of games played by player $i$. When $i \neq j$, the term $x_i \cdot x_j$ can be non-zero only when there is a match between player $i$ and $j$, which has exact values, $i.e.$, 1 and −1 are multiplied together and summed for every game, resulting in $-n_{ij}$.

Let $y$ be the vector of point differentials, where the $t^{th}$ component of $y$ is the point difference in the $t^{th}$ game, and $p = X^Ty$. Since the Massey rating $r_{ms}$ can be calculated from $Xr_{ms} = y$, then $X^TXr_{ms} = X^TXy$. Subsequently, the simple Massey linear equation is given as:

$$Mr_{ms} = p \tag{3}$$

Notice that the last row of $M$ is replaced by a vector of ones, and the last row of $p$ is replaced by zeros row because $M$ is s singular matrix, and Eq. (3) cannot be solved. Consequently, the addition of Massey scores of all players equals zero, $\sum_i r_{ms}(i) = 0$.

Notably, the win-loss method simply counts the number of wins and losses for each team without considering the margin of victory or defeat. It's a straightforward way to assess performance based on win-loss records. On the other hand, the Massey algorithm takes into account the margin of victory or defeat. It does not just treat all wins and losses equally. Teams are ranked based on a more sophisticated assessment of their performance, which can provide a more accurate representation of team strengths. In the intersection strategy, both Ni and nij (number of games played and number of games won) are equal for all genes. The rating is primarily based on the win-loss record, which is similar to the win-loss method. It focuses on the number of games won and lost without considering the margin of victory or defeat. In contrast, the union strategy assesses performance based on the percentage of games won. This strategy gives more weight to how convincingly teams win games, considering the margin of victory. It can provide a more fine-grained evaluation of team strength, rewarding teams that not only win but also do so decisively. Key distinction lies in how the algorithms handle the margin of victory or defeat. The win-loss method and intersection strategy primarily focus on the number of wins and losses, whereas the Massey algorithm and union strategy consider the margin of victory, providing a more nuanced and accurate assessment of team performance.

### *Colley's least squares method*

*Colley (2002)* discovered a ranking model that applies Laplace's rule of succession in a linear model. Let $W_i = \sum_j w_{ij}$ and $L_i = \sum_j l_{ij}$ be the number of wins and losses for team $i$.

First, the Colley matrix has a high connection with Massey matrix, $C = M + 2I$, where $I$ is

the $N \times N$ identity matrix, or it can be defined as $C_{ij} = \begin{cases} N_i + 2 & \text{if } i = j \\ -n_{ij} & \text{if } i \neq j \end{cases}$.

Let $b_i = 1 + \dfrac{W_i - L_i}{2}$ be the difference between the number of wins and losses, which is derived from the modified winning percentage $\dfrac{W_i + 1}{N_i + 2}$ to start the rating at 0.5. The derivation of $b_i$ from the modified winning percentage is presented in Data S1. Then, the rating $r_c$ is computed by solving the equation

$$Cr_c = b \tag{4}$$

Although, the Colley and Massey algorithms stem from different motivations (*Devlin & Treloar, 2018*), the two matrices are similar. Hence, they have led to nearly the same results since our application considers the score of matches as a win-loss record in Massey.

### Keener's method

*Keener (1993)* proposed a eigen-based ranking model by applying the Perron Frobenius eigenvector. The concept behind constructing the Keener matrix is to differentiate between a dense number of players with a win probability around 0.5. This model uses a non-linear skewing function, $h(x) = 0.5 + 0.5 \left( sgn(x - 0.5) \sqrt{|2x - 1|} \right)$. Next, the probability of winning, based on Laplace's rule of succession, is represented as $a_{ij} = \dfrac{W_{ij} + 1}{W_{ij} + W_{ji} + 2}$. Then, the Keener matrix is defined as $K_{ij} = h(a_{ij})$. Definitively, the Keener rating $r_k$ is obtained by solving:

$$Kr_k = \lambda r_k \tag{5}$$

where $\lambda$ is the largest eigenvalue, and $r_k$ is the corresponding eigenvector.

As mentioned, the intersection strategy considers the same number of games for all genes. The rating result is based on the winning percentage, similar to Colley, but it has the advantage of distinguishing near-zero probability, which can lead to more accurate results than Massey and Colley. For the union strategy, more candidate genes are considered; however, Keener requires more time to solve for eigenvectors and eigenvalues.

### Elo's ranking method

*Elo (1978)*'s system was first established for ranking chess players. Player rankings are based on their previous performances, changes in ratings occur through iterations. Elo's rating for team $i$ can be computed as:

$$r_E^{new}(i) = r_E^{old}(i) + f\left(\kappa_{ij} - \mu_{ij}\right) \tag{6}$$

where

$$\kappa_{ij} = \begin{cases} 1 & \text{if player } i \text{ win player } j \\ 0 & \text{if player } i \text{ loss player } j \\ 0.5 & \text{if player } i \text{ and player } j \text{ are tie} \end{cases} \quad \text{represents the actual outcome of the game.}$$

The constant $f$ is set to 10, and $\mu_{ij} = \dfrac{1}{1 + 10^{\left[r_E^{old}(j) - r_E^{old}(i)\right]/400}}$ is the expected probability of player $i$ winning against player $j$, constructed using a logistic function. Elo's method requires the initial ratings of each player as input; this work uses equal values for all players, with $r_E^{old} = 1500$. When considering the first pair of players, the player who wins the game gains a higher rating, while the loser's rating decreases. The process is iterative until the last pair of players is considered.

### Markov method

The Markov chain can be used for ranking by considering a voting process, where the stronger alternative is voted for by the weaker alternative (*Von Hilgers & Langville, 2006*). Nowadays, the Markov chain has been developed in various forms; for example, the (1,∞) variant is proposed to reduce the sensitivity of the model (*Vaziri, Yih & Morin, 2018*). Generally, the original Markov method is constructed using the (0,1) voting matrix

$$V_{ij} = \begin{cases} 1 & \text{if player } i \text{ win player } j \\ 0 & \text{otherwise} \end{cases}.$$

Next, the transition probability matrix $P$ is obtained by normalizing the voting matrix or dividing each element by its row summation. Then, the Markov rating vector $r_{mk}$ is obtained by solving:

$$r_mk = Pr_{mk}. \tag{7}$$

The Markov ranking method takes into account both the opponents and their level of strength. However, this method has displayed sensitivity to small changes in data and also requires a long computational time, making it more suitable for solving problems with small number of players.

### PageRank method

PageRank was first proposed by Google's founders, Larry Page and Sergey Brin, to rank web pages (*Brin & Page, 1998*). Unlike the previous methods, this model is constructed using a network. First, a directed graph $G$ is constructed by considering each node as a player, with directed edge pointing from the losing player to the winning player. A higher number of in-degrees for a node indicates a stronger opponent. Let $B_u$ be the set of neighboring nodes pointing to $u$, and $|v|$ be the outgoing degree from node $v$. Then, the PageRank score $r_p(u)$ of player $u$ is defined as:

$$r_p(u) = \sum_{v \in B_u} \frac{r_p(v)}{|v|} \tag{8}$$

Alternatively, the power method is applied to quickly solve for the PageRank rating. Let $A$ be the $N \times N$ adjacency matrix of the graph $G$. $A$ is normalized by row summation to obtain $Z$, and $G$ is a Google matrix computed as $G = \alpha Z + (1 - \alpha) E$, where $E$ is an

entirely $1/N$ matrix, and $\alpha$ is a damping factor, usually set to 0.85. Thus, the PageRank score $r_p(u)$ can be computed as:

$$r_p = r_0 G^c \tag{9}$$

where $r_0$ is the initial vector, which is set to $1/N$ if no initial vector is provided, and $c$ is the number of iterations needed to reach convergence. To apply PageRank in the union strategy, genes in $S_{up}$ are treated as nodes in the network for the up-regulated case, and $S_{down}$ for down-regulated case. Genes that participate many games and frequently win against strong opponents tend to have high PageRank scores.

### *Bi-directional PageRank method*

The improved model of PageRank is developed for sport ranking (*Zhou et al., 2022*). Bi-directional PageRank (BiPageRank) considers both win and loss whereas PageRank computes only the winning score. This work shows the outperforming results of BiPageRank when compared with PageRank using both synthetic data and application of four sports: soccer, basketball, ice hockey, and baseball. The BiPageRank can be computed as:

$$r_s = r_p - r_q \tag{10}$$

where $r_p$ is the PageRank score, and $r_q$ is the backward propagation of PageRank score, which can be calculated as $r_q(u) = \sum_{v \in Q_u} \dfrac{r_q(v)}{|v|^{in}}$, where $Qu$ is the in-neighbor of node $u$, and $|v|^{in}$ is the in-degree of node $v$. PageRank $r_p$ assigns a higher score to players who frequently win against strong teams. In contrast, $r_q$ assigns a higher value to players who lose to low-rated teams. Thus, the BiPageRank score improves upon PageRank by considering both the wins and losses of the players.

## Validation technique

Leave-one-out cross-validation (LOOCV) is applied to obtain the performance. For each iteration, one dataset is considered as a testing set, whereas the remaining ones are the training set. This process is performed on all datasets. To evaluate the performance, area under the ROC curve (AUC) and under the precision-recall curve (AUPR) are used as the measurement tools. The positive set is defined as genes with $logFC > 1$ and $adj.p.val < 0.05$ in the up-regulated case, and genes with $logFC < -1$ and $adj.p.val < 0.05$ in the down-regulated cases. Note that, in LOOCV, the normalized AUPR (AUPRN) is applied instead of AUPR because of the imbalance datasets. The AUPRN is computed from raw AUPR divided by baseline positive proportion (number of positive/total number of samples).

## Gene expression data

We applied two datasets, Hypertrophic cardiomyopathy (HCM) and Microarray Quality Control (MAQC), to validate the performance of our integration strategies.

*Gene expression data of hypertrophic cardiomyopathy*

Different experimental types, sample origins, and platforms used to collect data on HCM provide varying information. To ensure the reliability of results and confirm the importance of genes to the disease, it is crucial to include a larger number of samples in the model. In this study, we gathered nine datasets of HCM gene expression data collected from the Gene Expression Omnibus (GEO) database. The data comprise four microarray datasets: GSE36961, GSE32453, GSE68316 (*Yang et al., 2015*), and GSE1145 and five RNA-Seq datasets; GSE89714, GSE130036 (*Liu et al., 2019*), GSE160997 (*Maron et al., 2021*), GSE180313 (*Ranjbarvaziri et al., 2021*), and GSE141910. In total, there are 464 samples, consisting of 213 cases and 251 controls. The characteristics of the HCM gene expression data are provided in Table S1.

*Microarray quality control project*

The United States Food and Drug Administration (FDA) provides data of Microarray Quality Control (MAQC) and Sequencing Quality Control (SEQC). MAQC was first developed to evaluate agreement across microarray data and is provided in GSE5350 (*Li et al., 2014*; *MAQC Consortium, 2006*; *Su et al., 2014*; *Wen et al., 2010*). With the emergence of next-generation sequencing technologies, SEQC was introduced to access RNA-Seq performance, and it is available in in GSE56457 (*MAQC Consortium, 2014*), GSE47774 (*Su et al., 2014*), and GSE48016 (*Munro et al., 2014*; *Wang et al., 2014*). From the four types of samples provided by the United States Food and Drug Administration (FDA), we have selected two types of RNA samples: A (Universal Human Reference RNA) and B (Human Brain Reference RNA). We gathered nine datasets from GEO database to obtain gene expression data from 1442 samples, with 721 samples for each type, as provided in Table S2.

# RESULTS

After introducing the online platform, the results of various ranking techniques for HCM and MAQC are analyzed.

## Online platform

The integration of multiple gene expression datasets with GeneCompete can be accessed through https://genecompete.streamlit.app/. GeneCompete requires CSV input files of the gene expression table, with the first column containing gene names. This data can be pre-processed using any suitable tools for flexibility. The numerical column is also user-defined, with *logFC* applied as default.

As depicted in Fig. 2A, users need to specify the regulation case and strategy they wish to use. When selecting a union strategy, it's important to properly adjust the *logFC* threshold, as processing many genes can be computationally intensive. Before ranking, datasets are filtered with *logFC > thres* for up-regulation and *logFC < -thres* for down-regulation. We recommend keeping the number of candidate genes below 10,000 for user validation.

A

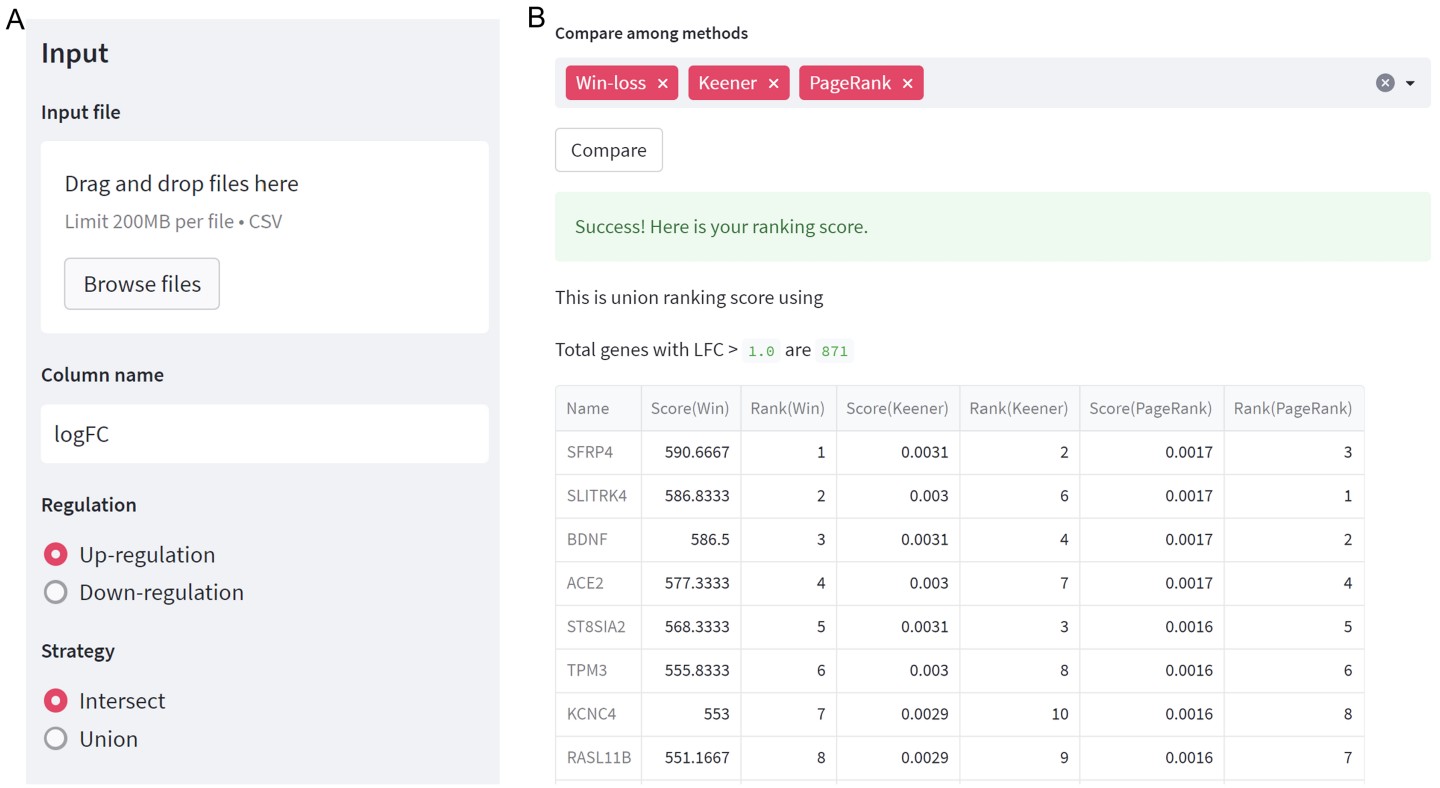

**Figure 2 GeneCompete: a web-based tool.** (A) The starting page for setting up an input and options. (B) The result page of selected ranking scores.

**Table 1 The total number of genes in each dataset.**

| No. | GEO accession no. | Number of genes | Number of genes with *logFC > 0* | Number of genes with *logFC < 0* |
|-----|-------------------|-----------------|----------------------------------|----------------------------------|
| 1 | GSE36961 | 37,846 | 20,152 | 17,694 |
| 2 | GSE32453 | 11,696 | 5,460 | 6,236 |
| 3 | GSE68316 | 6,768 | 2,490 | 4,278 |
| 4 | GSE1145 | 21,753 | 12,155 | 9,598 |
| 5 | GSE89714 | 15,240 | 9,320 | 5,913 |
| 6 | GSE130036 | 16,779 | 4,021 | 12,758 |
| 7 | GSE160997 | 13,758 | 2,668 | 11,090 |
| 8 | GSE180313 | 15,802 | 12,041 | 3,761 |
| 9 | GSE141910 | 17,124 | 8,612 | 8,512 |
| | Total | 45,695 | 36,715 | 32,461 |

Then, users can choose the ranking technique(s) they prefer, including Win-loss, Massey, Colley, Keener, Elo, Markov, PageRank, or Bi-PageRank. The example demonstrates a comparison of three methods: Win-loss, PageRank, and Keener. In Fig. 2B, the obtained results of the rating scores and rankings can be downloaded.

## Differentially expressed genes from multiple datasets of HCM

Differential expression analysis conducted on datasets obtained from different platforms can yield varying sets of important genes. Using more datasets can enhance the accuracy of gene identification. In this study, we incorporated a larger number of datasets compared to previous research (*Janyasupab, Suratanee & Plaimas, 2022*), resulting in an intersection of 3,194 genes, as opposed to the previous 3,259 genes. However, the presence of genes in all datasets does not necessarily indicate their significance in the disease context. Table 1 provides the total number of genes in each dataset, with a union of all datasets resulting in 45,695 genes. Hence, the advancement of GeneCompete within this research lies in our capacity to larger datasets, thereby furnishing a valuable tool for others to apply various ranking techniques to their own datasets. Additionally, the incorporation of a union strategy to handle multiple datasets enhances robustness and extents the list of potential gene candidates. To streamline computational efficiency, we have categorized these genes into two cases: up-regulated (with $logFC > 0$), yielding 36,715 candidate genes, and down-regulated (with $logFC < 0$), resulting in 32,461 candidate genes for ranking purposes. To show sensitivity of $logFC$ threshold, the absolute of $logFC$ is applied as a competing score ($|logFC| < thres$, $thres = 1, 2, 3, 4$). In Fig. S2, AUC and AUPRN tends to be lower in a higher $logFC$ threshold except for classical method and average $logFC$ which not consider the number of datasets.

### *Prioritization techniques with up- and down-regulation genes*

LOOCV involves leaving one dataset as the testing set while combining the others using ranking methods. Higher performance indicates a stronger relationship with the DEGs of the test set.

To compare the ranking performance with the original method, we consider two cases: up-regulation and down-regulation. The original or classical method for identifying DEGs based on applying specific criteria: a $logFC > 1$ and an *adj.p.val.* $< 0.05$ for up-regulation genes, and a $logFC < -1$ and an *adj.p.val.* $< 0.05$ for down-regulation genes. Subsequently, each gene's count score is calculated by summing the total number of datasets meeting the criteria of $logFC > 1$ and $logFC < -1$, for each up or down cases. The average $logFC$ (*Avg_logFC*) is then directly computed as the mean $logFC$ across datasets. The ROC curve becomes visible after a single iteration of leave-one-out cross-validation. The ROC curves for union strategy in up-regulation and down-regulation can be found in Datas S2 and S3, while the results for intersections are presented in Datas S4 and S5. Fig. 3 shows that the original method exhibits the low predictive power for DEGs in both up-regulation and down-regulation cases. The count score method improves upon the original approach, suggesting that genes present in more datasets with a high absolute $logFC$ are more likely to predict DEGs. Consequently, ranking scores that consider the number of datasets can be valuable for the prediction.

Notice that many cases demonstrate the union strategy yielding higher performance than the intersection approach across many ranking methods. In the case of the win-loss method, the union strategy, which considers the number of datasets a gene has joined, demonstrates better performance than the intersection strategy, with the same number of
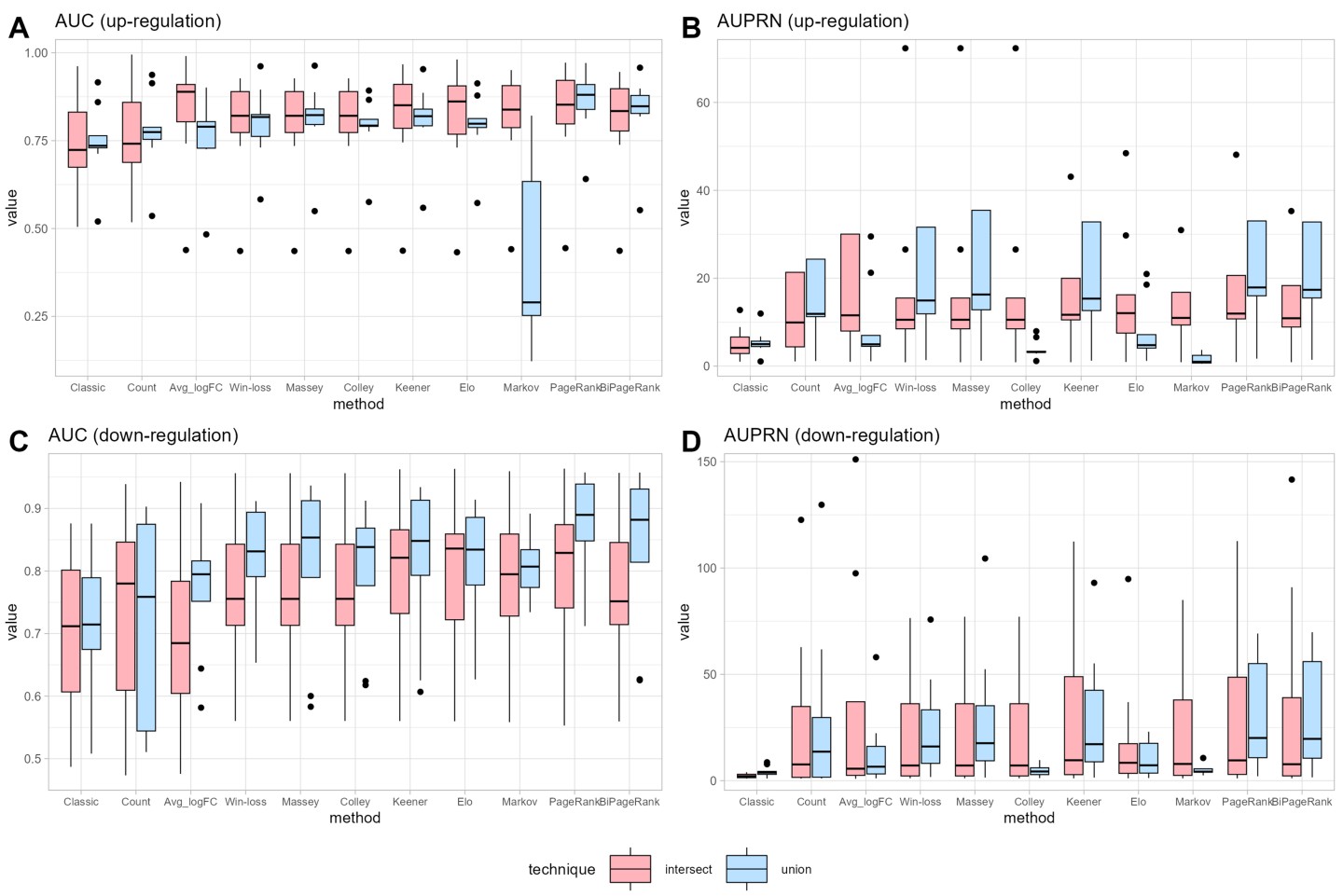

**Figure 3 Performance measurement.** (A) and (B) represent the performance in terms of area under the ROC curve (AUC) and under the precision-recall curve (AUPR) for up-regulated cases, respectively. (C) and (D) represent the performance in terms of area under the ROC curve (AUC) and under the precision-recall curve (AUPR) for down-regulated cases, respectively.

datasets. Massey's approach applied the winning score as input, showing similar performance for both methods. Colley and Elo, utilizing a probability of win, can reduce the effectiveness of the union ranking strategy due to genes that appear in a greater number of datasets having a higher probability of score reduction, unlike the intersection strategy that maintains a consistent score regardless of dataset count. A modification of the Keener matrix did not improve rankings for this task and achieved a similar performance to the win-loss method. The Markov method shows high sensitivity in union ranking, especially in cases of up-regulation. This implies that minor data changes can lead to substantial ranking differences; for instance, a lower-ranked player defeating the highest-ranked player might result in a significant increase in the former's ranking. Both PageRank and BiPageRank exhibit similar behaviors, though BiPageRank displays slightly lower performance. By the concept of PageRank, genes that wins other important genes tend to have a higher score. Moreover, we observe that genes that are presented in a higher

**Table 2  Top 10 genes detected by PageRank.**

| No. | Up-regulated genes | Down-regulated genes |
|---|---|---|
| 1 | SLITRK4 | FCN3 |
| 2 | SFRP4 | CORIN |
| 3 | CA3 | HOPX |
| 4 | FRZB | MYH6 |
| 5 | MXRA5 | SERPINA3 |
| 6 | SMOC2 | TUBA3E |
| 7 | THBS4 | CD163 |
| 8 | FNDC1 | SMTNL2 |
| 9 | FMOD | CCL2 |
| 10 | DIO2 | RARRES1 |

number of datasets play a higher number of matches and have a higher chance to receive a PageRank score. Among PageRank, Win-loss, Keener, and BiPageRank, as illustrated in Fig. 3, most instances demonstrate that PageRank and BiPageRank exhibit superior performance in terms of both the AUC of the ROC curves and the AUPR of the precision-recall curves. The common thread shared by PageRank, Win-loss, Keener's Algorithm, and BiPageRank is their focus on ranking genes within their respective relationship networks. While PageRank and BiPageRank underscore the significance of connections and links to other nodes, Keener's Algorithm takes into account both local and global influences from others. Conversely, Win-loss simplifies ranking by relying on binary outcomes in competitive scenarios.

It is worth noting that in both the intersection and union strategies, genes can be categorized based on their positive and negative *logFC* values. Comparisons between the separated and non-separated versions can be found in Fig. S1. Interestingly, the outcomes appear more favorable for the separated approach. However, it is important to highlight that the separated approach yields only 23 candidate genes for up-regulation and 105 for down-regulation.

### Winner genes with the best top ranking

The PageRank method method stands out as the best approach for intersection strategy. Tables S3 and S4 present the top 10 ranking genes for both up-regulated and down-regulated cases. The *logFC* values of genes in many datasets are lower than 1 in up-regulated case and greater than −1 in down-regulated case. This suggests the gene expression differences in HCM are not reaching a two-fold changes compared to normal patients.

Tables S5 and S6 display the top 10 ranking genes obtained by each method. Interestingly, the top genes identified by the win-loss, Keener, PageRank, and BiPageRank methods are quite similar. In Table 2, our method identifies the top-ranking genes using PageRank, which is supported by existing literature evidence. HCM is closely associated

with dilated cardiomyopathy (DCM), a type of heart muscle disease that can lead to heart failure and life-threatening arrhythmia (*Tobita et al., 2018*).

Among the top 10 up-regulated genes, SLITRK4 has been identified as a promising biomarker for HCM gene tests (*Zheng et al., 2021*). It is commonly differentially expressed in the datasets related to the study of HCM patients, such as GSE130036 and GSE36961 (*Cui et al., 2022*). SFRP4 is known to be involved in cardiac development and various cardiovascular diseases (*Zeng et al., 2019*). It has been identified as a hub gene in the HCM key module (*Ma et al., 2021b*) and as an up-regulated DEGs in HCM (*Ren et al., 2016*). In addition, SFRP4 is associated with ischemic cardiomyopathy, a type of heart muscle disease (*Alimadadi et al., 2020*), and has been verified as a hub gene associated with heart failure (HF) (*Zhou et al., 2020*).

For CA3, an increase in its expression has been confirmed by immunohistochemistry as a myocardial protein (*Coats et al., 2018*). Moreover, CA3 expression levels were significantly higher in the plasma of heart failure patients than in control patients (*Su et al., 2021*). FRZB has been identified as a hub gene in the HCM key module (*Ma et al., 2021b*) and hub biomarkers for dilated cardiomyopathy (DCM) (*Fang et al., 2022*). In addition, FRZB has been recognized as a potential immune-related key genes involved in ischemic cardiomyopathy through random forest analysis and nomogram (*Zheng et al., 2023*).

MXRA5 has been identified as a key gene with prognostic value in left-sided HF (*Zhou et al., 2020*). It is extracellular-associated proteins included in the top 500 genes in the HF consensus signature (*Ramirez Flores et al., 2021*). SMOC2 has been defined as a real hub gene of HCM due to its high intramodular connectivity values (*Jiang et al., 2021*). The protein encoded by the differentially expressed methylated gene SMOC2 was found to be upregulated in Chagas disease cardiomyopathy (*Shi et al., 2022*). THBS4 is implicated in severe HCM and heart failure pathogenesis (*Tsoutsman et al., 2013*). It is also predicted to play a role in the development of DCM (*Zhao et al., 2018*). THBS4 expression has been associated with hypertrophic cardiac disease (*Peisker et al., 2022*). FNDC1 was among the 10 most up-regulated transcripts in patients undergoing repair of tetralogy of Fallot heart tissue, compared with right ventricle donor tissue (*Brayson et al., 2020*). Both FNDC1 and MXRA5 have been identified as novel extracellular matrix (ECM) biomarkers in calcified valves, making them potential targets in the development and progression of aortic stenosis (*Bouchareb et al., 2021*).

FMOD has been identified as upregulated DEGs in heart failure (*Kolur et al., 2021*). It is a type of fibromodulin that is upregulated in clinical and experimental heart failure (*Andenæs et al., 2018*). DIO2 is a direct transcriptional target of the FoxO1 protein, which is involved in relative hypertrophic growth of neonatal cardiomyocytes *in vitro* and *in vivo* (*Ferdous et al., 2020*). It has been reported that DIO2 is up-regulate in the hearts of DCM mice (*Wang et al., 2010*).

For the top 10 of down-regulated genes, FCN3 is a key dysfunctional gene. It was identified by studying the network of differentially expressed genes between HCM and healthy controls (*Cui et al., 2022*). Additionally, FCN3 is associated with the development of HF (*Jiang, Zhang & Zhao, 2022*). CORIN was identified as a downregulated mRNAs in the myocardial tissues of patients with HCM (*Cao & Yuan, 2022*). It was reported to be a

cardiac protease that activates natriuretic peptides, the expression of which has been examined and studied in the activity of mouse and human failing hearts (*Chen et al., 2010*). Regarding HOPX, the relationship between HOPX gene variations and HCM was investigated. The results suggest that HOPX may cause pathogenesis or manifestation of HCM (*Güleç et al., 2014*). A study showed that HOPX expression is reduced and completely absent in severe heart failure (*Trivedi et al., 2011*). In addition, the HOPX gene plays an adjusted role in HCM pathogenesis through SRF-dependent genes (*Alkanli & Ay, 2019*). It was reported that the expression of MYH6 is dominant in human cardiac atria and plays roles in cardiac muscle contraction, including the composition of the cardiac muscle thick filament (*Razmara & Garshasbi, 2018*). Moreover, MYH6 mutations were evaluated in HCM phenotypes (*Hsieh et al., 2022*). The study showed that mutations in the MYH6 gene result in the abnormal development of cardiac muscle cells, which can lead to HCM.

SERPINA3 is significantly perturbed in heart failure proteins shared between two studies (*Chen et al., 2022*). It was reported to be downregulated in HCM compared to healthy controls (*Chen et al., 2018*). It is a common down-regulated DEGs in GSE130036 and GSE36961 (*Cui et al., 2022*). TUBA3E was identified as an HCM hub gene in the negative module (*Jiang et al., 2021*). It is also a common down-regulated DEGs in GSE130036 and GSE36961 (*Cui et al., 2022*). TUBA3E is included in a list of down-regulated genes expressed in patients with both HCM and DCM (*Chaffin et al., 2022*).

A study suggested that the potential function of CD163 macrophages is in supporting the homeostasis of cardiac tissue (*Zhang et al., 2021*). CD163 plays a key role in the pathogenesis of HCM (*Zhao et al., 2016*). It is a common down-regulated DEGs in GSE130036 and GSE36961 (*Cui et al., 2022*). A study reported that SMTNL2 is a down-regulated HF gene (*Kolur et al., 2021*). Regarding CCL2, it was reported that the CCL2-CCR2 signaling pathways are associated with the development and progression of cardiovascular disease (*Zhang et al., 2022*). RARRES1 expression was observed to be absent in the HCM samples in many of the fibroblast populations (*Larson et al., 2020*). It is one of the top three DCM down-regulated genes (*Ma et al., 2021a*).

To confirm the biological relevance of each ranking, we performed a gene set enrichment analysis. The results of up-regulation and down-regulation are shown in Tables S7 and S8, respectively. For up-regulation, genes in each method are involved in similar pathways, such as the extracellular region (GO:0005576), extracellular region part (GO:0044421), extracellular matrix (GO:0031012), extracellular space (GO:0005615), proteinaceous extracellular matrix (GO:0005578) and neurogenesis (GO:0022008). For down-regulation, genes were enriched in immune response (GO:0006955), extracellular region (GO:0005576), and defense response (GO:0006952).

## Differentially expressed genes from MAQC datasets

We also investigated the performance of GeneCompete by applying data from Microarray Quality Control (MAQC). In Fig. S3, we present the performance results of each method based on both intersection and union approaches. Notably, the classical method demonstrates the weakest performance among all methods evaluated. When comparing

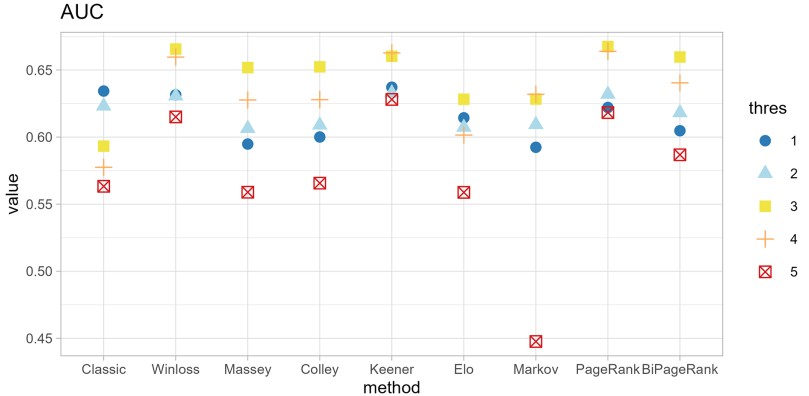

**Figure 4** **Area under the ROC curve of different methods.**

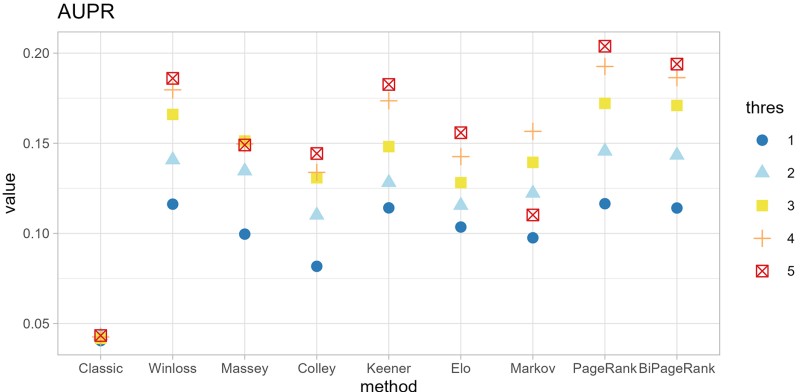

**Figure 5** **Area under the precision-recall curve of different methods.**

the AUC values, the union algorithm consistently yields lower scores compared to the intersection method across all ranking methods. However, the AUPRN values are notably higher when utilizing the Win-loss, Massey, Keener, PageRank, and BiPageRank methods. One limitation to consider is that the intersection algorithm considers only 415 genes as players, whereas the union approach includes 22,988 genes in the up-regulation case and 17,971 in the down-regulation case. Consequently, we opt for the union strategy due to its ability to maintain similar performance even when dealing with genes that exhibit significant differences. In the union strategy, PageRank emerges as the method with the highest average performance, as depicted in Table S9.

First, the pre-processing steps are performed in 'limma' package. In this part, we applied the absolute value of *logFC* as the input for the competing score. Genes with a higher absolute value of *logFC* can be either expressed more highly in sample A or B. The results are compared by using all provided methods with several *logFC* thresholds (*thres*). The performances are validated using TaqMan quantitative PCR technology (*MAQC Consortium, 2014*). These 1,044 gene symbols were obtained using 'seqc' library in R.

**Table 3 The top-ranking hits for different thresholds.**

| No. | Thres = 1 | Thres = 2 | Thres = 3 | Thres = 4 | Thres = 5 |
|---|---|---|---|---|---|
| 1 | GFAP | GFAP | GFAP | GFAP | GFAP |
| 2 | ALB | AHSG | ALB | ALB | ALB |
| 3 | AHSG | ALB | HBE1 | SPARCL1 | SPARCL1 |
| 4 | STMN2 | HBE1 | STMN2 | STMN2 | GPM6A |
| 5 | AFP | STMN2 | AHSG | HBE1 | HBE1 |
| 6 | HBE1 | AFP | AFP | GPM6A | SYT4 |
| 7 | APOA2 | PMEL | PMEL | AFP | STMN2 |
| 8 | GPM6A | APOA2 | APOA2 | SYT4 | HBB |
| 9 | PMEL | GPM6A | SPARCL1 | HBB | SYNPR |
| 10 | HBZ | SPARCL1 | GPM6A | APOA2 | AFP |

The original method filters genes by the condition of $|logFC| < thres$ and $adj.p.val < 0.05$. The ranking methods also filter with five thresholds ($thres = 1, 2, 3, 4, 5$) before calculating scores. The results in Figs. 4 and 5 show that similar AUC values are obtained from all methods, while the original method yields the worst AUPR. This indicates that the ranking can improve the performance in predicting SEQC. Among the methods, PageRank shows the highest performance, especially in AUPR. It is followed by Win-loss, Keener, and BiPageRank, which have similar ranking performance.

Furthermore, 'GeneCompete' also requires a $logFC$ threshold when the union strategy is selected. We presented here the different threshold selection with the corresponding performance of all ranking methods as shown in Figs. 4 and 5. Most AUPR decreases when the threshold is lower, whereas the AUC for each method is not dependent on the threshold. Table 3 shows that the top-ranking hits found by PageRank at each threshold produce similar genes. For example, GFAB, ALB, GPM6A, and HBE1 occur in the top 10 ranking of all five thresholds. In addition, many genes in the top 10 ranking are also found in the TaqMan list, indicating the high predictive performance of PageRank. Genes verified by TaqMan were underlined in Table 3.

## The computational cost of each ranking technique

Our online platform, GeneCompete, is designed for gene expression data ranking analysis and integration. It encompasses various ranking algorithms, each with distinct computational characteristics in terms of differences in time and cost of calculation. In our approach, the technique is notably based on the number of genes that overlap or combine for competitive analysis. Our exploration, involving different gene counts and datasets, reveals that most algorithms offer reasonable computational costs, ensuring swift results as depicted in Fig. 6. However, an increase in the gene count corresponds to extended computational time, particularly evident in the case of the Markov and Elo methods. The computational cost of Elo and Markov exhibits exponential growth with higher gene counts. Under such circumstances, Elo's method showcases the lowest performance due to the iterative nature of both Markov and Elo, which involve repetitive calculations until

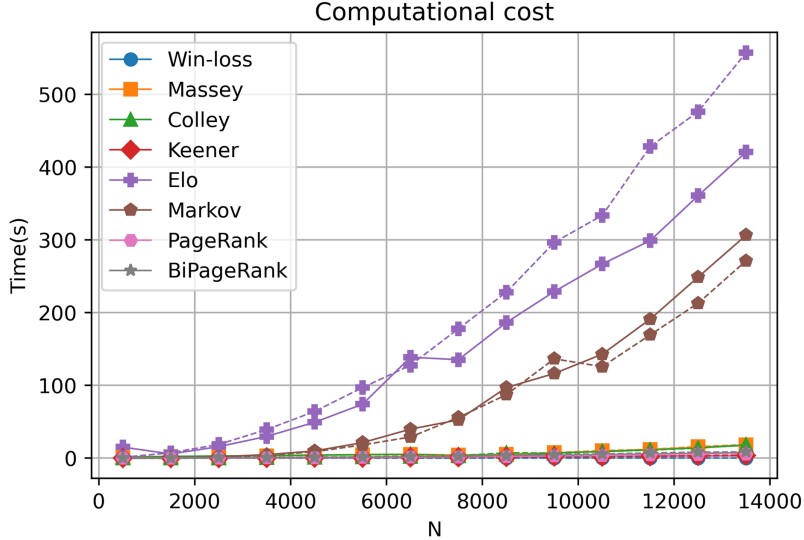

**Figure 6 Computational cost.** A dashed and solid lines represent intersection and union strategies, respectively.

stability is achieved. Notably, PageRank and BiPageRank demonstrate favorable outcomes in both identifying crucial genes (winning genes) and maintaining reasonable computational costs.

## DISCUSSION

Nowadays, transcriptomics data have significantly increased due to technological advancements. Analyzing heterogeneous data plays a vital role in merging information from diverse sources and platforms. Larger volumes of data provide stronger evidence regarding the correlation between genes and diseases, making it crucial to consider integration techniques. This study specifically focuses on combining gene expression data from various datasets and platforms. Numerous research studies have aimed to develop techniques for obtaining log-fold change, which directly indicates the contrast in expression between normal and diseased patients. Integration of various data from different platforms provides more complete information to cope with a disease of interest to better understand genes functions based on their expressions. This underscores the importance of developing distinct tools for analyzing differential expressions. However, most of the algorithms have been designed for individual datasets. In our study, we leverage ranking techniques to merge multiple expression datasets and prioritize the most relevant genes for diseases.

For this task, ranking methods are employed as a novel concept for obtaining ranking scores. In this concept, genes are treated as players, and their log-fold change values serve as scores. The number of datasets utilized represents the number of matches. In this model, *p*-values are not incorporated, and high-ranking genes tend to have lower *p*-values, indicating the significance of differential gene expression. The results demonstrate that the union strategy outperforms the intersection strategy. This is because the set of genes that appear in all datasets alone cannot determine significance. Consequently, by considering

the union of genes across all datasets, the gene pool expands, resulting in improved utility. However, using a large number of genes is not suitable for certain models, particularly linear equations. Hence, the union strategy initially segregates positive and negative log-fold change values to facilitate their utilization in the up-regulated and down-regulated models, respectively. This work faces a limitation concerning the selection of the *logFC* threshold. When opting for a low threshold, it can lead to an overwhelming number of genes, making it difficult to discern the most relevant ones from noise. Conversely, a very high threshold results in a limited gene selection, potentially overlooking important candidates with slightly lower *logFC* values. Striking the right balance between sensitivity and specificity in threshold selection is crucial for obtaining meaningful results.

Among all ranking methods employed, PageRank demonstrates the most predictive performance in terms of both AUC and AUPR. The PageRank algorithm leverages both the strength of the player (gene) and the strength of the opponent (high *logFC* genes) to determine the ranking scores. In the case of the union strategy, PageRank is also influenced by the number of datasets in which a gene participates. To provide further clarity, a gene receives a higher PageRank score if its *logFC* is greater than that of genes with high *logFC* and if it is involved in a larger number of datasets in the case of up-regulation. Conversely, for down-regulation, the opposite applies. This approach aims to identify genes that consistently exhibit significant differential expression across multiple datasets. In the result section, we also present the win-loss method, which closely aligns with the performance of PageRank. The similarity in the top-ranking genes between these two methods suggests that the win-loss method can be a viable alternative for ranking genes in this context.

In this study, we introduce GeneCompete, an online platform for conducting ranking analysis and integrating gene expression data. The input for this platform consists of a list of data frames representing the *logFC* table. Prior to analysis, pre-processing steps can be performed using various tools such as 'limma' (*Smyth et al., 2005*), 'DESeq2' (*Love, Huber & Anders, 2014*), and 'edgeR' (*Robinson, McCarthy & Smyth, 2010*). It is worth noting that certain datasets may exhibit high absolute *logFC* values, which can result in a large number of candidate genes during the selection of positive and negative cases. As shown in Fig. 6, a higher number of genes leads to higher computational time, especially for Markov and Elo's methods. To address this, larger filtering thresholds can be implemented to reduce the number of genes.

This approach is not restricted to the specific disease studied in this research; it can be extended to various other diseases as well. Our method is versatile, employing the calculation of each ranking algorithm without the need for disease-specific information or computations. To apply this methodology, one simply adapts the input data to suit the relevant disease and particulars. The algorithm then autonomously computes ranking scores, organizes genes based on these scores, and offers outcomes for subsequent analysis and experimentation. Users can easily leverage the method, opting for the most suitable technique and identifying top genes of interest by utilizing scores generated by GeneCompete's diverse algorithms. Varied pre-processing techniques can be employed for data from different platforms, and managing a substantial gene count is achievable through the application of appropriate filtering thresholds. Notably, the PageRank

technique, in conjunction with the union strategy, is highly recommended due to its computational efficiency and impressive ranking performance.

## CONCLUSIONS

This study introduces a novel online tool, called 'GeneCompete', that combines a union strategy with various ranking approaches to integrate multiple gene expression datasets. The effectiveness of these algorithms is demonstrated through their application to HCM and MAQC gene expression data obtained from microarray and RNA-Seq technologies. Not only can genes with their log-fold change scores from expression analyses be used in this tool, but other types of data containing lists of genes with their scores can also be input. GeneCompete will automatically summarize the ranking scores and prioritize the genes based on their competition scores, as well as identify the overall winner for the competitions.

The union strategy is proposed as it considers a larger pool of candidate genes compared to previous integration pipelines. The ranking scores exhibit strong performance, particularly with the PageRank method, in both up-regulation and down-regulation cases. Notably, the top-ranking genes tend to have high absolute log-fold change values in individual datasets, indicating their potential biological significance.

The promising results obtained from this work suggest that in the future, it will be possible to develop more accurate prioritization techniques for identifying important genes. These techniques could significantly contribute to advancements in gene expression analysis and facilitate the identification of key genes associated with various biological processes and diseases.

## ACKNOWLEDGEMENTS

The authors acknowledge the NSTDA Supercomputer Center (ThaiSC) for providing computing resources for this work.

### Funding

This research was funded by the National Science, Research and Innovation Fund (NSRF), and King Mongkut's University of Technology North Bangkok with Contract no. KMUTNB-FF-66-08. The funders had no role in study design, data collection and analysis, decision to publish, or preparation of the manuscript.

### Grant Disclosures

The following grant information was disclosed by the authors:
National Science, Research and Innovation Fund (NSRF).
King Mongkut's University of Technology: KMUTNB-FF-66-08.

### Competing Interests

The authors declare that they have no competing interests.

## Author Contributions

- Panisa Janyasupab conceived and designed the experiments, performed the experiments, analyzed the data, performed the computation work, prepared figures and/or tables, authored or reviewed drafts of the article, and approved the final draft.
- Apichat Suratanee conceived and designed the experiments, analyzed the data, performed the computation work, authored or reviewed drafts of the article, and approved the final draft.
- Kitiporn Plaimas conceived and designed the experiments, analyzed the data, performed the computation work, authored or reviewed drafts of the article, and approved the final draft.

## Data Availability

The data is available at NCBI GEO: GSE36961, GSE32453, GSE68316, GSE1145, GSE89714, GSE130036, GSE160997, GSE180313, and GSE141910 for HCM dataset; and GSE5350, GSE56457, GSE47774, and GSE48016 for MAQC dataset.

The code is available at GitHub and Zenodo:

- https://github.com/panisajan/GeneCompete.

- panisajan. (2023). panisajan/GeneCompete: Initial (Initial). Zenodo. https://doi.org/10.5281/zenodo.8383849

Our web-based program is available at: https://genecompete.streamlit.app/.

## Supplemental Information

Supplemental information for this article can be found online at http://dx.doi.org/10.7717/peerj-cs.1686#supplemental-information.

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
