# Peer review of "GeneCompete: an integrative tool of a novel union algorithm with various ranking techniques for multiple gene expression data"

_PeerJ Computer Science, doi:10.7717/peerj-cs.1686_

## Round 0.1 · original submission · Major Revisions

The reviewers have substantial concerns about this manuscript. The authors should provide point-to-point responses to address all the concerns and provide a revised manuscript with the revised parts being marked in different color.

**Language Note:** The review process has identified that the English language must be improved. PeerJ can provide language editing services - please contact us at copyediting@peerj.com for pricing (be sure to provide your manuscript number and title). Alternatively, you should make your own arrangements to improve the language quality and provide details in your response letter. – PeerJ Staff

Reviewer 1 ·

Basic reporting

The paper is very well-written, well-structured and provides detailed descriptions of the proposed methodology and its evaluation.

Experimental design

The experimental design and data are enough to explain the objective of the paper.

Validity of the findings

The innovativeness is moderate.

Additional comments

In this manuscript, the authors summarized and discussed a novel union algorithm for multiple gene expression. On this basis, they tested GeneCompete, a web-based tool on the expression datasets of HCM and MAQC project. This work provides new insight and opinion into developing of the gene expression analysis. The manuscript is well-organized and clearly stated. I would suggest accepting it after the following minor concerns are addressed.
1. What are the similarities and differences between the intersection and union integrating processes?
2.What were the similarities and differences in the ranking performance among PageRank, Win-loss, Keener, and BiPageRank?
3.How can this methodology be applied to other diseases, and how can the high number of genes be effectively managed?

Reviewer 2 ·

Basic reporting

In the manuscript under review, Janyasupab and colleagues have developed an innovative web-based tool, named "GeneComplete," that aggregates genes from diverse datasets to rank their expression levels. Their proposed 'union' strategy is an impressive method, allowing all the genes to be effectively ranked. The authors further explored a variety of ranking algorithms and evaluated the performance of both 'intersection' and 'union' strategies.

The introduction of GeneComplete could offer significant contributions to future studies on differentially expressed genes (DEGs), providing a comprehensive platform for analysis. Nevertheless, there are certain aspects of the manuscript that still raise concerns.

Experimental design

My primary concern lies in the design and implementation of the 'intersection' and 'union' strategies. The authors argue that the drawback of the intersection strategy is the potential omission of certain genes not present in every dataset. I propose that the authors explore a combined approach of these two strategies. For instance, they could aggregate all datasets in the manner of the intersection strategy while incorporating all genes as in the union strategy, rather than keeping the datasets separate. The rankings could then be normalized based on the number of matches for each gene, mirroring the method used in the union strategy.

Validity of the findings

Additionally, I find myself concerned about the section on performance comparison. In line 407, the authors state, "The union strategy yields higher performance than the intersection in most ranking methods." However, Figure 3 seems to depict two distinct trends in AUC scores when dealing with up-regulated cases versus down-regulated ones. It's crucial that the authors provide more precise and accurate descriptions of their measurements. The existing claims should be carefully evaluated in the light of the visual data presented.

Additional comments

Furthermore, I noticed some linguistic revisions are required. Specifically, in line 150, the word 'loss' should be replaced with 'beat' for clearer context. In line 160, it appears the authors intended to say, 'The number of games between each pair of genes is determined by how many times the two genes appear in the same dataset.'

Moreover, the presentation of the algorithms could be improved for clarity. In the W_matrix formula presented in both algorithms, the 'j' indicator is absent for the second 'gene'. This should be corrected for an accurate representation. In the case of the union algorithm, concerning the down-regulation part (line 6), it seems it should read 'logFC < -thres'?

I recommend a thorough scrutiny of the manuscript to eliminate any remaining grammatical errors and inaccuracies in descriptions. A careful review can help enhance the overall clarity and precision of the content.

Reviewer 3 ·

Basic reporting

The English language should be improved to be clear, grammarly correct, logically fluent, and technically correct. Some examples are:
1. Clarity: line 58-59; line 87 “those methods”; line 88 unclear about what was “qualified”; line 144 provide a simple explanation of “regulation cases” will be helpful
2. Grammar: line 80 “refereed”; line 94 “is proposed” -> “has been proposed”; line 99 “was improved” -> “has improved”.
3. Logic: line 81 “However, …, but …”; line 92-100 not easy to follow – consider reordering the methods based on their similarity and/or time of publication.
4. Technical: line 130-135 need more statistical rigor, for example, we typically don’t “accept” the null; the p-value is the probability of falsely rejecting the null while the null is true; a p-value close to 0 means the difference is unlikely observed by chance but doesn’t necessarily mean the difference is greater; the p-value itself is not subject to false discovery rate, the problem is caused by multiple comparisons.
5. Other: avoid subjective words like “obviously” in line 86; I personally feel “disease games” in line 89 is not very appropriate, maybe simply say “diseases”.

The introduction section showed the context, and the referenced papers are relevant. A few related questions/comments are:
1. The authors seem to indicate that GeneCompete is specifically designed to integrate RNA-seq and microarray data. However, the method itself should be applicable to any analyses that involve ranking by some statistics. Therefore, I recommend the authors consider a broader range of datasets and applications.
2. Please confirm if “data integration” is the most appropriate terminology for this paper. It may be confused with algorithms that try to remove unwanted differences and align different datasets, such as Harmony for single-cell RNA-seq.
3. Line 66-68 states there are strengths and weaknesses of RNA-seq and microarray, and it’s beneficial to combine them. It would be more convincing if the authors provide a brief explanation of the strengths and weaknesses and the rationale for combining the two.
Figure quality needs to be improved. For example, figures in general have low resolutions. Figure 4 & 5 legend “group” is not clearly defined.

Experimental design

The research question is well-defined, relevant, and meaningful. The dataset and methods are described, but some important details need to be added/clarified:
1. GeneCompete computes ranking scores based on multiple rankings obtained from differential expression analyses of multiple datasets. Therefore, it is important to clearly explain how the ranking algorithms were extended to accommodate k matches, which is missing in part or completely for most of the methods.
2. Line 338-340: the GSE accession numbers should be provided instead of GPL number.
3. In the results and discussion section, the authors briefly explained why the union strategy was performed for up-regulated and down-regulated genes separately. However, I think it should be explained in the methods section because this is a very important difference between the union and intersect strategy. Moreover, the intersection method with up/down genes separated should also be included in the comparisons of the results, even just as supplementary materials.
4. Algorithm 1
a. Line 3: why not use T_list[0]?
b. Line 4: no need to intersect with T_list[0] again.
c. Line 13 & 15 are incomplete
d. Line 13-19: is it the same as re-ordering the columns and rows of strictly upper/lower triangular matrix by increasing/decreasing order of the logFC (upper triangular matrix for win, lower for lose; increasing order for down-regulation, decreasing for up-regulation)?
5. Algorithm 2
a. Line 9: Data_FC not defined
b. Line 10-13: why not simply initiate Union_set as an empty set?
c. Line 21-28: see comments for Algorithm 1 Line 13-19.
6. Some notations/wordings to be clarified
a. Line 197: S_t is the set of genes, not number of genes
b. Line 200: S_up and S_down were defined for each dataset before taking the union according to Algorithm 2, but here the order is reversed. If the text is correct, please clarify how the “logFC_i” was computed across k datasets.
c. Line 227: shoud “Xtt” be “Xti”? In the definition of “Xti”, who is the opponent of team i when you determine whether i won or lost the game?
d. Line 229: Ni has not been defined. Please show the derivation of Mij.
e. Line 246: please derive bi from the modified wining percentage.
f. Line 271: consider using other notations instead of “K” and “Sij” since “k” and “S” have been used above.
g. Line 301: consider using other notation instead of “S” since it has been used above.
7. Line 124, explain why to choose the highest p-value. Choosing the lowest seems more reasonable if the goal is to avoid leaving out interesting genes (consistent with the union strategy).
8. Line 238-240: comparison between the two strategies is not very impressive given that ranking by the proportion of wins is equivalent to the number of wins for the intersection strategy.

Validity of the findings

I commend the authors for evaluating their performances with two big datasets, each containing thousands of samples from several sub-datasets. I have a few comments/questions regarding the results section.
1. I test the web-tool at 9:44 PM (UTC-4) on July 24, 2023. The application failed when I try to use default settings for example data 1, 3, and 4. To test the robustness of the application, I selected all the methods except for win-loss and directly clicked “compare”. The application is still offline by the time this review is submitted.
2. Line 408: according to the median statistics in Figure 3, the union strategy is only her than the intersection in 3/9, 5/9, and 5/9 cases in panels A, B, and D. I think the conclusion for “most” methods is too strong.
3. The limitations of the study should be included in the discussions.
4. Figure 3 shows the AUROC and AUPRC for different integration and ranking methods. I would recommend also showing additional data.
a. Show representative ROC curves or report the partial AUROC/AUPRC, because the high sensitivity and high specificity part is more relevant than the whole curve.
b. Since the interpretation of AUPRC is dependent on the baseline proportion of positive cases, and the baselines are different for each test data, it is not very helpful to show all the raw AUPRCs. Showing normalized statistics, such as the ratio/difference between AUPRC and baseline positive proportion, may be helpful.
c. As the authors mentioned in the introduction and methods section, the computational burden varies across different integration strategies and ranking methods. Therefore, a comparison of computational cost and scalability is desired.
d. In this and MAQC data analysis, consider also including very simple and naïve ranking strategies, such as ranking genes in each dataset by logFC and then taking the average or median of the ranks, using p-value instead of logFC, computing average or median logFC or p-value and rank by the average or median, etc.
e. Sensitivity analysis for the choice of logFC threshold like in Figures 4 and 5.
5. In the analysis of the MAQC dataset, the authors only showed the results of union + PageRank. However, given this valuable dataset has TaqMan-validated ground truth, a more comprehensive evaluation like in Figure 3 should be performed.
6. Line 427-431: how many genes overlap with top-ranking genes identified by other integration strategy and ranking method combinations? Performing a gene set enrichment analysis may be useful to compare the biological relevance of top-ranking genes identified by different method combinations. Also, are there specific reasons why these four genes were highlighted on line 430?
7. Line 516: the statement is not fully correct since thresh = 5 has the lowest AUROC based on Figure 4.
8. Line 520-521: please explicitly point out (e.g. bold face or underline in the table) which genes were verified by TaqMan.
9. Line 396-401: Is the “counting score” the same as the “original”? If yes, the conclusion in Line 398-401 conflicts with that in line 396-398. If not, please include data for “counting score” in the figures. While the authors concluded the original method “receives the lowest performance”, it doesn’t seem to be the worst in all cases.
10. Line 416-417: unclear what “sensitivity” means. It would be helpful to provide some insights into why Markov performed so badly for the union strategy according to Figure 3A.

Additional comments

This paper presents GeneCompete, a tool to rank genes based on their differential expression statistics in multiple datasets using different ranking strategies. The authors explained the method with text and figures and evaluated its performance using two groups of datasets. The experiment design and findings seem valid, but some method details are missing, and the presentation and interpretation of results need to be improved. In addition, the performance of the union strategy compared to intersection seems limited according to the presented results, and the advancement compared to (Janyasupab, 2022) needs to be further accessed.

---

## Round 0.2 · accepted · Accept

Reviewers are satisfied with the revisions and I concur to suggest accepting the revised manuscript.

Reviewer 2 ·

Basic reporting

The authors have significantly improved the manuscript from its original version. I am satisfied with the revisions and recommend its acceptance for publication.

Experimental design

I have no additional concern over the experimental design of the manuscript.

Validity of the findings

The findings are valid.